# Styloid Jugular Nutcracker: The Possible Role of the Styloid Process Spatial Orientation—A Preliminary Morphometric Computed Study

**DOI:** 10.3390/diagnostics13020298

**Published:** 2023-01-13

**Authors:** Giorgio Mantovani, Pietro Zangrossi, Maria Elena Flacco, Giovanni Di Domenico, Enrico Nastro Siniscalchi, Francesco Saverio De Ponte, Rosario Maugeri, Pasquale De Bonis, Michele Alessandro Cavallo, Paolo Zamboni, Alba Scerrati

**Affiliations:** 1Department of Translational Medicine and for Romagna, University of Ferrara, 44121 Ferrara, Italy; 2Department of Neurosurgery, University Hospital of Ferrara, 44121 Ferrara, Italy; 3Department of Environmental and Preventive Sciences, University of Ferrara, 44121 Ferrara, Italy; 4Department of Physic and Earth Science, University of Ferrara, 44122 Ferrara, Italy; 5Division of Maxillofacial Surgery, BIOMORF Department, University of Messina, 98122 Messina, Italy; 6Department of Experimental Biomedicine and Clinical Neurosciences, School of Medicine, Postgraduate Residency Program in Neurological Surgery, Neurosurgical Clinic, AOUP “Paolo Giaccone”, 90127 Palermo, Italy; 7Vascular Diseases Center, University Hospital of Ferrara, 44121 Ferrara, Italy

**Keywords:** Eagle Jugular Syndrome, jugular stenosis, venous hypertension, styloid process

## Abstract

Styloid Jugular Nutcracker (SJN, also known as Eagle Jugular Syndrome EJS) derives from a jugular stenosis caused by an abnormal styloid process, compressing the vessel in its superior portion (J3) against the C1 anterior arch. It could be considered a venous vascular variant of Eagle Syndrome (ES). Main clinical features of this ES variant are headache, pulsatile tinnitus and dizziness, possibly related to venous hypertension and impaired cerebral parenchyma drainage. In our opinion, conceptually, it is not the absolute length of the styloid bone that defines its abnormality, but its spatial direction. An elongated bone pointing outward far away from the midline could not compress the vein; vice versa, a short styloid process tightly adherent to the cervical spine could be pathological. To prove this hypothesis, we developed a semi-automatic software that processes CT-Angio images, giving quantitative information about distance and direction of the styloid process in three-dimensional space. We compared eight patients with SJN to a sample of healthy subjects homogeneous for sex and age. Our results suggest that SJN patients have a more vertically directed styloid, and this feature is more important than the absolute distance between the two bony structures. More studies are needed to expand our sample, including patients with the classic and carotid variants of Eagle Syndrome.

## 1. Introduction

Styloid Jugular Nutcracker (SJN), also known as Styloidogenic-cervical spondylotic internal jugular venous compression or Styloid-induced internal jugular vein stenosis, is an emerging pathological entity variously related to several central nervous system disorders. 

It could also be considered as a vascular venous variant of Eagle Syndrome (Eagle Jugular Syndrome EJS [1]), a well-known, but not so well understood, illness in Ear–Nose–Throat (ENT) surgical departments.

The central focus of Eagle Syndrome (ES) is the stylohyoid complex, which is composed of styloid process, stylohyoid ligament and the lesser horn of the hyoid bone. The styloid bone starts from the inferior portion of the temporal bone, just medially to the base of the mastoid process, laterally to the jugular foramen, and directs inferiorly, medially and anteriorly, passing anteriorly and laterally to the C1 anterior arch and transverse process to direct into the infratemporal fossa. From its tip, the stylohyoid ligament further directs anteriorly and inferiorly to attach onto the hyoid bone lesser horn. Three muscles originate from the tip of the styloid process: styloglossus (innervated by hypoglossal nerve), stylohyoid (facial nerve) and the stylopharyngeal (glossopharyngeal nerve). 

Embryologically, the stylohyoid complex is derived from Reichert’s cartilage, which arises from the second pharyngeal arch, but its exact origin is still matter of debate [2]. Its ossification starts at the end of pregnancy and continues during the first 8 years of life [3].

Several developmental abnormalities and alterations in bone homeostasis have been advocated to contribute to the styloid process elongation or, generically, alteration: the presence of two ossification centers in the embryological styloid bone, embryonic mesenchymal conversion to osteoid matrix, osteoarthritic changes and diseases of calcium-phosphate maintenance with heterotopic ossification (e.g., Paget’s disease) [4,5,6,7].

The classic variant of ES was first described in 1652 by the Italian surgeon Pietro Marchetti (labeled as intermittent respiratory distress) [8], but the full description of the syndrome is attributed to Dr. Watt W. Eagle in 1937 [9] as an association of dull, constant and nagging pain variably referred to face and neck, dysphagia, odynophagia and sensation of a foreign body in the throat and otalgia, exacerbated by yawning and swallowing. Symptomatology usually starts after a trauma such as road accident, neck manipulation, accidental falls, tonsillectomy or dental extraction [8]. Rarely, dysgeusia and inspiratory/expiratory stridor have been reported [10,11]. Its precise diagnosis could be quite challenging due to clinical heterogeneous presentation and unspecific symptoms [12]. Additionally, its differential diagnosis is wide including glossopharyngeal neuralgia, occipital neuralgia, sphenopalatine neuralgia, temporomandibular disorders, dental infection, tonsillitis, mastoiditis, and migraine [2]. Several hypotheses have been made on its pathogenesis, mainly referable to post-traumatic reactive hyperplasia or metaplasia or anatomical variance stretching on the sensory nerve ending in the parapharyngeal region [2]. 

Soon after, a vascular variant of ES had been described by Dr. Eagle themself, referred as “stylo-carotid artery syndrome”, caused by an elongated styloid process impinging the carotid artery and associated nerves. Clinical features consist of pain in the parietal region of the skull or in the superior periorbital region, and an increased risk of cerebro-vascular ischemic accidents: transient visual loss, syncope, Horner’s syndrome, arterial dissection, transient ischemic attack, stroke and even sudden death [13,14,15,16]. 

Treatments for both the classic and carotid variants are usually conservative in first attempt with the employment of gabapentin, amitriptyline, valproate, carbamazepine, image-guided corticosteroid injections and antiplatelet agents [17]. Surgical indication is given for non-respondent symptomatology or higher risk of cerebro-vascular damage. Styloidectomy is the preferred option and could be performed by intraoral access (pro: no skin incision, potentially shorter operative time; cons: higher risk of infection, incomplete styloid removal) or by latero-cervical access (pro: minimal generation of airway edema, ability to perform bilateral styloidectomy in the same sitting, cons: higher risk of cranial nerves damage) [2].

In the third, and less known, variant of ES the conflict arises between the styloid process and the anterior arch of C1 in its anterior–lateral aspect. In this passage, the Internal Jugular Vein (IJV), exiting from the skull through the jugular foramen along with the cranial nerves from IX to XII, is compressed between the two bony structures in its superior segment (J3). 

Clinical depiction of this entity is far less defined, but the most frequent symptoms are headache (not specifically facial pain and not frequently present in the other two variants), tinnitus (frequently pulsatile), insomnia, visual disturbances (such as blurred vision) and hearing impairment [1]. It has also been recently related to cerebral venous sinuses thrombosis [18]. Up to 50% of patients present a bilateral IJV stenosis without preference for sex, mean age at onset is 38.6 years [19].

Recently, cerebral venous drainage is gaining increasing interest because its association with various neurological conditions, such as Alzheimer’s disease, Multiple sclerosis and Meniere’s disease, related to Chronic Cerebrospinal Venous Insufficiency (CCSVI) [20]. To date, it is clear that CCSVI is the final emerging phenomena coming from a wide range of anatomical and pathological conditions. 

In addition, all SJN symptoms could be reasonably linked to an obstructed cerebral venous outflow, subsequent venous hypertension and impaired parenchymal drainage.

In their mathematical model of cerebral venous outflow, Gadda et al. [21] demonstrated that a severe stenosis in the IJV could significantly affects the venous sinuses pressure, causing venous hypertension in the cortical vessels.

Consistent with that, SJN has been directly associated to non-aneurismatic subarachnoidal hemorrhage (na-SAH), also known as sine materia SAH [22], a pathology classically correlated to venous anatomical variations and impaired outflow, particularly regarding the Basal vein of Rosenthal or occluded cerebral venous sinuses [23,24].

Differently from the classic and carotid variants of ES, only 1/3 of patients with EJS present an effectively elongated styloid process [25], even if definition of “normal” length is still debated, varying from 1.5 to 4 cm in various studies [26,27,28]. By consensus, a styloid process longer than 30 mm could be seen as a risk factor for ES development [2]. 

Indeed, a longer styloid bone confers greater movement capacity at its distal end, increasing the risk of adjacent neurovascular compromise. In a physiological range of movement, during a neck extension the tip of the styloid process shows around 30 degrees of angular motion, and considering a styloid process length of 30 mm, then the tip movement measures about 15 mm [29,30]. 

Anatomically the internal jugular vein is antero-lateral to the styloid process. In the case of SJN, intuitively, the length of the bony process could not be the cause of IJV compression, but its spatial orientation could be. Indeed, even a shorter process that is in close adherence to C1 could have the possibility to determine significant stenosis. In fact, Ho et al. [25] in their series reported that, in patients with a unilateral IJV compression determining SJN, the side of stenosis was the one in which styloid process was shorter. Previous papers on this topic provided no quantitative data about styloid process spatial orientation, especially compared to healthy subjects.

This is the first study, to our knowledge, providing quantitative data about styloid process spatial orientation in patients suffering from Eagle Jugular syndrome and a control group of healthy subjects.

To prove that the styloid process spatial orientation could be a key feature in developing SJN, we developed a dedicated semi-automatic software and retrospectively analyzed patients’ images at our institution. 

## 2. Materials and Methods

We collected cervical CT-angiography (CTA) images from 8 patients with SJN confirmed by venous angiography. SJN was radiologically defined as a reduction in IJV caliber at least of 80% on axial cuts compared with the normal vein proximal to the stenosis, according to Jayaraman [31]. Then a control group was selected: each patient was matched with a control subject of same sex and age, who underwent CT-angiography for another reason, and in which no evidence of styloid anomalies or IJV obstruction were found (see Table 1).

Patients gave a written consent for the use of anonymized clinical and radiological data. The local ethical committee gave the authorization for the study (693/2019/Oss/AOUFe). Images were anonymized by random number association and again randomly mixed.

The 3D Slicer^®^ [32] was used by a blind operator to create 3D segmentation on CTA (originating a .stl extension file), depicting for each patients the two styloid processes (right and left) and C1. A threshold automatic selector was applied to better define the bony borders, connecting points with the same Hounsfield density (see Figure 1). 

Then, we developed a dedicated software using Anaconda [33] vers. 2-2.4.0 to calculate principal spatial orientation parameters of the styloid process (see Figure 2). The major axis of styloid was defined as the polyline connecting the barycenter of each axial section previously segmented. To describe the orientation in three-dimensional space we used CT predefined axis. Particularly in axial view, x-axis is defined proceeding latero-laterally (from left to right); y-axis postero-anteriorly (from occipital bone to nose), z-axis cranio-caudally (from head to feet) (see Figure 3). 

Comparing the styloid major axis to the orthogonal CT reference, the software calculates:-Angle on x-axis: the angle between the styloid major axis and the x-axis. Visually, it is the direction from lateral to medial. The bigger this angle, the lesser the medial direction of the styloid bone.-Angle on y-axis: the angle between the styloid major axis and the y-axis. Visually, it is the direction from back to nose. The bigger this angle, the lesser the anterior direction of the styloid bone.-Angle on z-axis: the angle between the styloid major axis and the z-axis. Visually, it is the direction from head to feet. The bigger this angle, lesser the inferior direction of the styloid bone.

## 3. Results 

Results are summarized in Table 2. We found a statistically significant difference in y-axis spatial orientation (*p* < 0.005): Eagle styloids presented a mean angle of 83.3° (sd 19.5), while normal styloids presented an angle of 66.1° (sd 8.40). Mean minimum and maximum distances between C1 and styloid processes were longer in Eagle patients than in controls (8.03 mm vs. 7.44 mm for minimum and 87.9 mm vs. 81.7 mm for maximum, respectively). Other measurements, such as minimal distance between styloid process and C1 anterior arch or x-axis direction (lateral-to-medial) did not show any statistical significance. 

## 4. Discussion

Results of this study show that the styloid process orientation on the Y-axis could be a possible factor causing the internal jugular vein stenosis: pathological styloid has a greater y-axis angle than control, that is to say they are more vertically directed. 

Indeed, causes and consequences of this kind of jugular stenosis are debated: relationships between extra-cranial veins and intra-cranial physiology are still poorly understood, and little is known about how a venous compression can result in a wide spectrum of clinical situations, from asymptomatic to severely ill patients. Particularly, regarding SJN, patients can be found with small styloid and severe jugular stenosis and vice versa.

In addition, a significant stenosis is not always associated with a symptomatic clinical syndrome. In this regard, Buch et al. [34] reported that IJV caliber presents wide variations in a healthy population undergoing CT scan, and thus caution should be taken in considering it pathological. We should consider, anyway, that they reported as significant a 50% reduction in IJV caliber. Similarly, Jayaraman et al. [31] reported that an extrinsic compression of the IJV superior segment is a common finding in unselected patients, and that severe stenosis is not always associated with collateral formation. 

Consistent with that vague clinic-radiological definition, SJN has been variously treated over time: anticoagulation is the most frequent prescription, followed by diuretics (acetazolamide), statins or antithrombotics [19]. However, the majority of patients do not have a good clinical result with conservative therapy. Styloidectomy with an external latero-cervical approach is the most commonly performed surgical procedure both as primary strategy or as rescue treatment after venous stenting failure [19]. As an alternative, the resection of the C1 arch has been performed, especially in cases in which a deep indentation of the lateral mass is detected at the CT scan [35]. Often, surgery and endovascular procedure need to be combined in order to effectively restore a functional IJV. Surgical manipulation could be insufficient to completely re-expand the chronically compressed IJV given its lack of muscular layers and low pressure flow; stent placement then allows one to mechanically restore the vessel diameter [36]. More than 70% of patients have some benefit from invasive procedures, even if long term follows up are missing in the literature. It seems that tinnitus, papilledema and visual disturbances are the most relieved symptoms, while headache and dizziness usually do not respond [19].

Probably, a deeper comprehension of SJN cannot rely only on radiological data. For example, a recent review found that an elongated styloid process can be found up to 30.2% of the general population, without correlation with sex, age or geography [37], but with a preference for bilaterality. Radiological information needs to be integrated with functional data regarding flow, pressure, direction and collateral pathway of the blood inside the IJV. 

Other kinds of IJV stenosis have been described, in relation to neurological illness. One that presents some similarities with SJN, in regard to the Central Nervous System (CNS) involvement, is the Jugular Entrapment and ventricular Dilation (JEDI) Syndrome [38]. A case of a woman presenting with signs and symptoms of intracranial hypertension (papilledema, headache) and hydrocephalus (Evan’s Index 0.36) has been described. Interestingly, the patient progressively developed pulsatile tinnitus, similarly to SJN patients. Cerebral FDG positron emission tomography (FDG-PET) revealed diffuse hypometabolism. Venous sinuses thrombosis has been ruled out. A bilateral external compression of the omohyoid muscle on the internal jugular veins was apparent at high-resolution B-mode neck ultrasound, never modified by posture or increase of intrathoracic pressure. During invasive intracranial pressure (ICP) registration, a surgical release of the IJV has been performed. Immediately after muscle transection, ICP fell about 6 mmHg. On the 24-month follow up, the patient was asymptomatic and a funduscopic examination showed resolution of the papilledema. A later FDG-PET detected an improvement in cerebral metabolism and brain MRI showed a decrease of brain ventricles (Evans index 0.31).

In a continuous spectrum of clinical entities, SJN have been also described in relation to intracranial hypertension, clinically manifested as headache exacerbated by neck flexion and successfully treated with styloidectomy [39,40,41]. Clinical and radiological overlap between various forms of ES are also described, thus adding complexity to the scenario [42].

A recent study built a 3D model derived from data of a patient suffering from SJN and analyzed how ICP changed in an elastic tube simulating IJV during head rotation, considering one IJV closed and the other one dynamically compressed by the styloid. They showed that ICP has a proportional increase with maximal axial rotation ipsilateral to the side of styloid-induced stenosis. This is partially in contrast with clinical presentation, in which headache is often exacerbated by contralateral head rotation [43]. Interestingly, the highest simulated ICP was observed when the styloid process was 50% resected [44].

Considering the anatomical position of the IJV stenosis, mathematical models could bring clues regarding their impact on cerebral circulation. Recently, it has been demonstrated that stenosis in the proximal portion of IJV (J3) has the greatest potential to impair outflow from the brain compared to more distal ones, and that jugular valve pathology can be exacerbated by flow disturbances evoked by IJV stenosis at this level [45]. These considerations, together with the clinical surgical outcome, provide indirect confirmation that SJN clinical depiction is mainly due to retrograde venous hypertension.

Our results suggest that styloid process’ spatial orientation could be a factor in the development of SJN: Eagle Jugular patients have a greater y-axis angle, that is to say, a more vertical styloid bone, compared to healthy subjects.

Interestingly, other axes are quite similar between the two groups, especially regarding the lateral-to-medial direction (x-axis).

Moreover, from a bony point of view, as we analyzed only the osseous CT-windows (see Figure 4) (250–1000 Hounsfield Units), the mean distance between styloid and C1 is smaller in healthy people, counter intuitively and unexpectedly. A possible explanation could be as follows: styloid bone is surrounded by fibro-muscular tissue and continues in the stylo-hyoid ligament, not visible in the mentioned bony windows. Compression can result hence from the connective tissues around a vertical directed styloid, more than from the bone itself. This is a limitation in our software, shared also with mechanical models.

This is a retrospective monocentric pilot study, with a limited sample of patients, and our software still has no external validations. To overcome some of these biases, images were analyzed by a single blind operator, and 3D segmentations were made using a semi-automatic threshold selector.

Our model considers a fixed head position, neutrally placed as for radiological protocol. Thus, it lacks data about styloid-C1 during head rotation, flexion and tilting.

To our knowledge, this is the first study to quantitatively analyze the morphometric features of SJN patients’ styloid processes in regard to 3D spatial orientation.

## 5. Conclusions

Our results suggest that a vertical styloid process is a key feature of SJN, more than the absolute distance between the latter and C1 anterior arch. Next steps will be to validate our software and expand the sample, including in the analysis patients with classic ES and the carotid variant, and integrating data of cerebrovascular flow dynamic variations, to ultimately describe this pathology in its whole complexity.

## Figures and Tables

**Figure 1 diagnostics-13-00298-f001:**
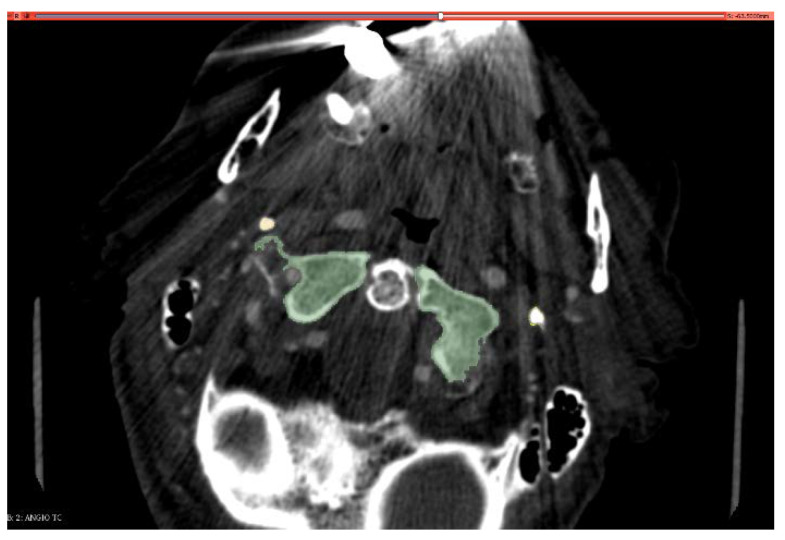
CTA segmentation using threshold selector.

**Figure 2 diagnostics-13-00298-f002:**
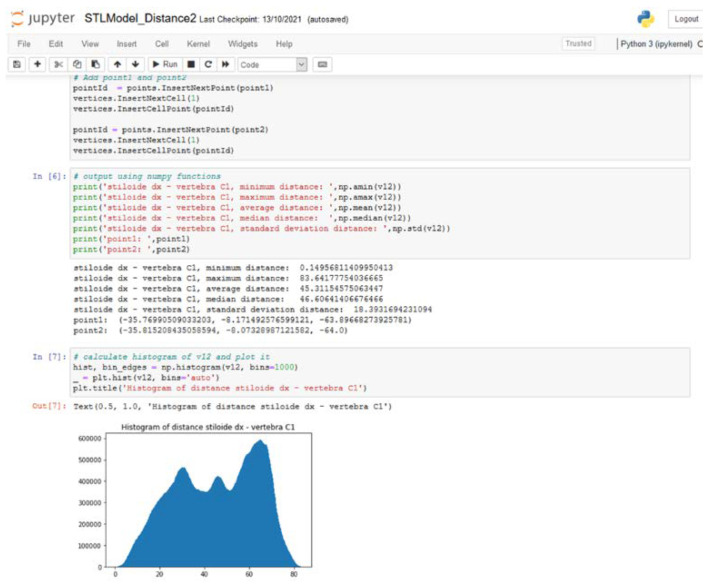
Dedicated software to calculate spatial orientation.

**Figure 3 diagnostics-13-00298-f003:**
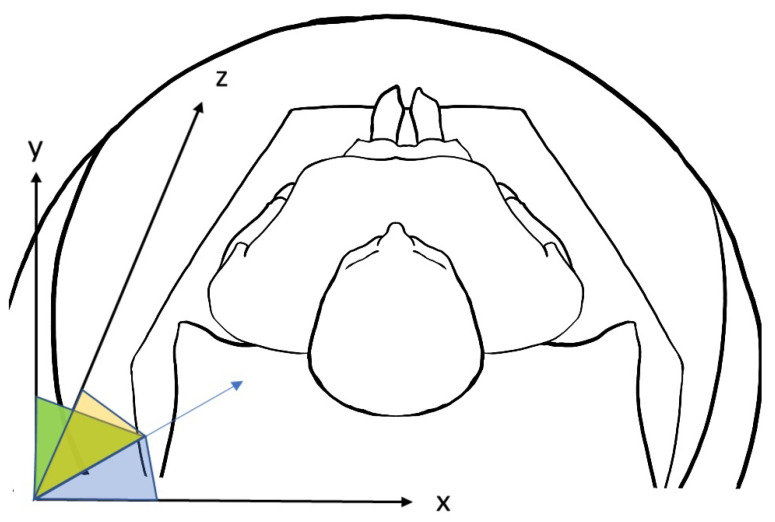
Spatial orientation of x, y and z axis of styloid process.

**Figure 4 diagnostics-13-00298-f004:**
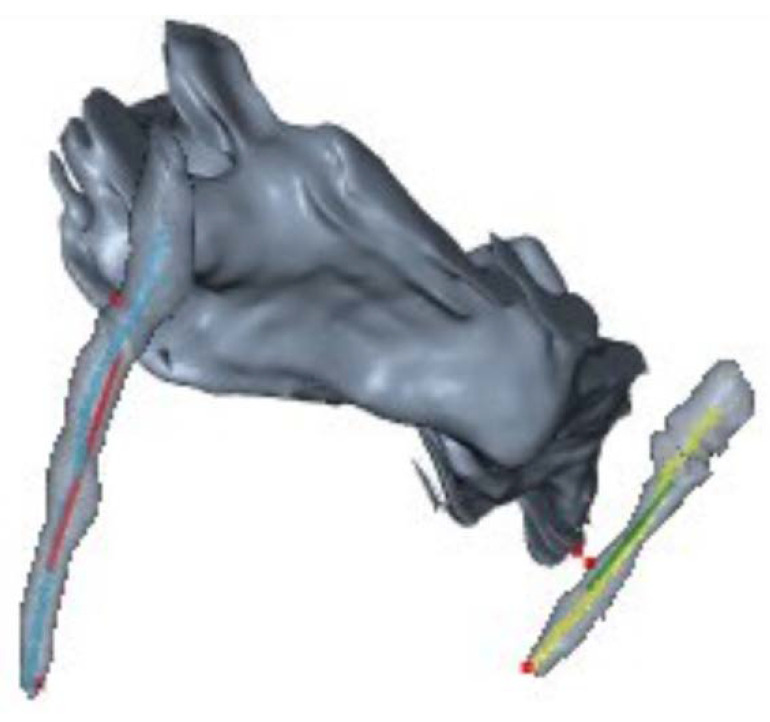
3D bony reconstruction.

**Table 1 diagnostics-13-00298-t001:** Demographic data.

	Sex (M/F)	Mean Age (M/F, years)
Case (n = 8)	4/4	73/61.5
Control (n = 8)	4/4	73/61.5

**Table 2 diagnostics-13-00298-t002:** Characteristics of the sample, overall and by Eagle syndrome status.

Variables	Overall Sample	Eagle Syndrome	No Eagle Syndrome	*p*-Value *
	(n=32)	(n=15)	(n=12)	
Styloid-C1 process (mm):				
Minimum distance, mean (SD)	7.72 (3.47)	8.03 (3.59)	7.44 (3.44)	0.7
Maximum distance, mean (SD)	84.7 (14.4)	87.9 (7.97)	81.7 (18.1)	0.5
Average distance, mean (SD)	45.2 (2.76)	45.3 (2.52)	45.2 (3.03)	0.9
Styloid-angle (degrees):				
x-axis, mean (SD)	90.7 (15.7)	90.6 (16.4)	90.7 (15.5)	0.9
y-axis, mean (SD)	74.2 (16.9)	83.3 (19.5)	66.1 (8.40)	0.005
z-axis, mean (SD)	26.9 (8.60)	24.9 (7.44)	28.7 (9.37)	0.2
Right-left styloid processes (mm):	(n=16)	(n=7)	(n=9)	
Minimum distance, mean (SD)	63.2 (12.4)	66.5(5.32)	60.7 (15.9)	0.8
Maximum distance, mean (SD)	89.7 (6.86)	91.7 (8.51)	88.2 (5.27)	0.6
Average distance, mean (SD)	77.7 (3.84)	78.5 (3.23)	77.1 (4.33)	0.6
* Kruskal-Wallis test; SD: standard deviation.				

## Data Availability

Not applicable.

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
