# Peer review of "Styloid Jugular Nutcracker: The Possible Role of the Styloid Process Spatial Orientation—A Preliminary Morphometric Computed Study"

_diagnostics, 2023, doi:10.3390/diagnostics13020298_

Round 1

Reviewer 1 Report

The authors have performed a preliminary study of the styloid process in individuals with the relatively rare condition of Eagle syndrome. The study is potentially valuable because it appears to provide a comparison of suffers with a control population, which does not appear to have any precedent in the published literature.

As the authors clearly understand and indicate, this is, in effect a pilot study, and the findings require confirmation from a more formal study. Even so, there is a significant flaw even for a pilot study because of the absence of detail of how the control group was selected: on Page 3, the control group is described as “random patients”, but later in the same sentence, this group is described as “homogenous for sex and age”. With such small numbers, a truly random selection of control patients is capable of providing substantial bias, so we need to know what is meant by “homogenous for sex and age”. Was there, for example, some form of stratified matching of test subjects with control subjects? Or if the control subjects are truly random, we should at least be provided with a simple table showing details of age and sex so that we can judge for ourselves whether there is any possibility of bias.

Background information about the Eagle syndrome is comprehensive and interesting, and both Introduction and Discussion are very detailed. The sections feel more like a review paper, and much of the content has no bearing upon the single positive finding of a difference in the orientation of the styloid process in one axis only.

The language is somewhat difficult to understand in this places, and there are idiosyncrasies that suggest word selection by a digital translation system. The authors might benefit from the assistance of a colleague with good understanding of medical English. For example, there are a number of errors in the first few paragraphs of the Discussion that might not be obvious to a non-medical editor:

-          “depicted” does not really make sense in this context, and I suspect the authors mean “understood“

-          “ouvert” should be “overt”

-          “It is to note” should be “It is of note”

-          “Coherently” would probably be better rendered as “Consistent”

-          I like the word “foggy”, but the word “vague” would be more conventional in this context

There are many similar discrepancies elsewhere in the paper: For instance, “venular” on Page 3 implies changes in venules, whereas I suspect that the word “venous” is meant. This

There are occasional medical terms not conventionally seen in English medical texts: “sine materia” is very rarely used, and we would simply describe the “basal vein of Rosenthal” as “basal vein”. However, these terms are very easily understood, so I do not feel strongly about whether they should be changed or not.

Reviewer 2 Report

I have the following concerns:

1. Literature review section can be added to provide a detailed background of the study.

2. Add the research gaps of the previous studies that your paper is addressing to contribute the novelty.

3. Results needs more explanation. Also, compare your results with existing similar studies to show its effectiveness.

Round 2

Reviewer 1 Report

I have no further comments.